# Causal Head Gating: A Framework for Interpreting Roles of Attention Heads in Transformers

**Andrew J. Nam**
Princeton Laboratory for AI
Natural and Artificial Minds
Princeton University
andrewnam@princeton.edu

**Henry C. Conklin**
Princeton Laboratory for AI
Natural and Artificial Minds
Princeton University
henry.conklin@princeton.edu

**Yukang Yang**
Department of Electrical and Computer Engineering
Princeton University
yy1325@princeton.edu

**Thomas L. Griffiths**
Department of Psychology
Princeton University
tomg@princeton.edu

**Jonathan D. Cohen**\*
Princeton Neuroscience Institute
Princeton University
jdc@princeton.edu

**Sarah-Jane Leslie**\*
Department of Philosophy
Center for Statistics and Machine Learning
Princeton University
sjleslie@princeton.edu

## Abstract

We present *causal head gating* (CHG), a scalable method for interpreting the functional roles of attention heads in transformer models. CHG learns soft gates over heads and assigns them a causal taxonomy—facilitating, interfering, or irrelevant—based on their impact on task performance. Unlike prior approaches in mechanistic interpretability, which are hypothesis-driven and require prompt templates or target labels, CHG applies directly to any dataset using standard next-token prediction. We evaluate CHG across multiple large language models (LLMs) in the Llama 3 model family and diverse tasks, including syntax, commonsense, and mathematical reasoning, and show that CHG scores yield causal, not merely correlational, insight validated via ablation and causal mediation analyses. We also introduce *contrastive* CHG, a variant that isolates sub-circuits for specific task components. Our findings reveal that LLMs contain multiple sparse task-sufficient sub-circuits, that individual head roles depend on interactions with others (low modularity), and that instruction following and in-context learning rely on separable mechanisms.

## 1 Introduction

Large language models (LLMs) [1, 2, 3] represent state-of-the-art systems across a wide array of domains, exhibiting remarkable generalization and problem-solving capabilities. Yet, as these models grow in scale and complexity, they become increasingly opaque, making it more difficult to understand, predict, or control their behavior, which raises concerns about safety and misuse [4, 5, 6]. This has motivated a growing body of work on *interpretability*, which seeks to better understand how LLMs learn and represent information, and how their responses can be shaped [7, 8].

---

\*Equal contribution; authors listed alphabetically

39th Conference on Neural Information Processing Systems (NeurIPS 2025).

Interest has focused in particular on transformer-based architectures [9] such as GPT [1], LLaMA [3], Gemma [10], and DeepSeek [2], in which the central processing blocks consist of multi-head attention followed by multi-layer perceptrons. Here, there has been considerable research on the roles of individual attention heads, which have been found to exhibit some level of human-interpretability [11, 12, 13].

Two broad categories of approaches dominate research on mechanistic interpretability in LLMs. The first uses a trained mapping from latent representations to human-interpretable concepts, such as syntactic features [7, 14, 15] or identifiable items (e.g., the Golden Gate Bridge [16]). The second uses causal interventions to identify portions of a single weight matrix or individual attention heads responsible for a specific behavior [17, 18]. These approaches often focus on small portions of a model, 'zooming in' [19] in an effort to interpret the role of a single computational subgraph. However, in deep-learning models, computation is often distributed [20] and the role of one component is dependent on another [21, 22, 23], making the behavior of such complex distributed systems difficult to predict from an understanding of their parts alone [24].

To apply a distributed perspective to mechanistic interpretability, we introduce *causal head gating* (CHG) which identifies a parametrically weighted set of heads that contribute to a model's execution of a given task. Given a dataset that defines a task, we fit a set of gating values for each attention head that applies a soft ablation to its output using next-token prediction, so that task-facilitating heads remain unaltered while any task-interfering heads are suppressed. Using a simple regularization procedure that further separates irrelevant heads from those that facilitate or interfere with task performance, CHG assigns meaningful scores to each attention head across an entire model according to its task contribution. We use these scores to define a taxonomy of task relevance according to how individual attention heads contribute to a model's distributed computation of a given task, describing each head as *facilitating, interfering* or *irrelevant*. In this respect, CHG offers an exploratory complement to standard hypothesis-driven approaches to mechanistic interpretability, assigning causal roles without relying on predefined hypotheses about what each head might be doing.

Beyond its conceptual contribution, CHG also offers several practical methodological advantages over existing mechanistic interpretability tools. First, because CHG operates directly on next-token prediction, it avoids the need for externally-provided labels [7, 14, 15, 16], controlled input-output pairs [7, 14, 15], or rigid prompt templates [25, 12, 13], which are often required for decoding and interventional approaches. Second, CHG naturally accommodates complex target outputs, including chain-of-thought reasoning [26], where the solution spans multiple intermediate steps. Finally, CHG is highly scalable: it introduces only one learnable parameter per attention head and requires no updates to the underlying model weights, so that the CHG parameters can be fitted in minutes using gradient-based optimization, even for LLMs with billions of parameters. Thus, in settings where analyzing complex dependencies between heads is important, it is feasible to fit large samples of CHG values to estimate a distribution over gating values in a bootstrap fashion.

To test its efficacy, we apply CHG across a diverse set of tasks—mathematical, commonsense, and syntactic reasoning—and across LLMs ranging from 1 to 8 billion parameters with varying training paradigms. We use CHG to analyze not only *where* specific computations take place, but also *how distributed* they are across attention heads, and how these patterns vary across different tasks and models. We also validate the causal scores produced by CHG by comparing them against targeted ablations as well as causal mediation analysis [12, 25], showing strong agreement between predicted and observed effects. Finally, we extend CHG to a contrastive setting to identify distinct sub-circuits that support instruction following versus in-context learning, suggesting that even semantically similar tasks can be underpinned by separable mechanisms.

Our main contributions are fourfold:

1. We introduce causal head gating (CHG), a parametric, scalable method for identifying potentially distributed, task-relevant sub-circuits in transformer models without requiring prompt templates or labeled outputs, and extend it with contrastive CHG to isolate heads supporting specific sub-tasks.
2. We propose a simple causal taxonomy of heads—facilitating, interfering, and irrelevant—that quantifies the effect of each on task performance using CHG-derived scores.
3. We use CHG to show that models contain multiple task-sufficient sub-circuits with varying degrees of overlap, suggesting head roles are not fully modular but depend on interactions with other heads.

4. We use CHG to show that instruction following and in-context learning rely on context-dependent separable circuits at the head level, where CHG-guided gating can selectively suppress one mode without substantially disrupting the other.

The accompanying repository for this paper can be found at `https://github.com/andrewnam/causal_head_gating`.

## 2 Related Work

**Representational decoders** Representational decoders are models trained to map hidden activations to externally labeled properties [7, 14, 15], estimating the mutual information between representations and those properties [27, 28]. However, such probing results are difficult to interpret: simpler decoders may underfit and miss relevant features (false negatives), while complex decoders may overfit and learn spurious correlations (false positives) [29, 30], requiring complexity-accuracy tradeoffs to contextualize results [30]. Moreover, although decodability indicates that a property is encoded in the representation, it does not imply that the model uses that information for its task, highlighting a correlational finding rather than a causal one [31]. Finally, representational decoders require labeled datasets, constraining their use to curated, predefined properties. For a comprehensive review of the probing framework and its limitations, see [27].

Sparse autoencoders (SAE) can be viewed as a related approach, where the autoencoder reconstructs representations through a sparse bottleneck to reveal modular or interpretable features [16, 32]. However, like probing classifiers, their insights remain correlational and still depend on post hoc labeling or interpretation, inheriting the same supervision bottleneck. In contrast, CHG performs direct interventions on model components without external supervision and proposes sufficient sub-circuits to the default unablated model, thereby identifying causal links between attention heads and model behavior on a task.

**Causal mediation analysis** Causal mediation analysis (CMA) [33, 34] is used to identify the functional roles of specific attention heads by crafting controlled prompt pairs that isolate a hypothesized behavior, then intervening on model components to measure their causal effect on outputs. For instance, in the indirect-object-identification (IOI) task [25], sentences like "When Alice and John went to the store, John gave a drink to..." are used to identify attention heads responsible for resolving coreference. By patching specific head outputs from a source sentence into a structurally matched target, and checking whether the model changes its prediction (e.g. "Alice" instead of "Mary"), CMA localizes the relevant circuit. It has also uncovered head-level roles in function tracking [12], symbol abstraction [13], and other structured settings [35].

However, CMA relies on manually crafted prompt templates and clear mechanistic hypotheses, which limits its scalability to more complex domains. In open-ended tasks like mathematical reasoning [36, 37, 38], the diversity of required knowledge makes it hard to design effective controlled inputs. A single shared template is unlikely to accommodate even two prompts from the MATH dataset [37], such as: "If $\sum_{n=0}^{\infty} \cos^{2n} \theta = 5$, what is $\cos 2\theta$?" and "The equation $x^2 + 2x = i$ has two complex solutions; determine the product of their real parts." Moreover, LLMs often solve such problems most effectively via chain-of-thought reasoning [26], which unfolds over multiple steps, further complicating the use of a unified prompt structure.

**Head ablations** Despite the use of multiple heads being commonplace in transformer-based architectures, it has been observed that multiple, and sometimes the majority of, heads can be entirely pruned with minimal impact on model performance [39, 18, 40, 41]. Moreover, entire layers can be pruned while retaining model performance [42, 43, 44]. However, existing works on pruning attention heads have focused primarily on custom-trained small-scale transformers [39, 18, 40] or BERT-based [45] models [41, 43], and the literature is limited for modern causal LLMs such as GPT [46, 1] and Llama [3].

Head pruning has also been used to validate findings from other interpretability methods, such as CMA [25, 13] or attention pattern analysis [18]. In these studies, researchers first identify heads believed to perform specific functions, then ablate them to test their causal impact. Such targeted ablations often lead to disproportionate drops in performance, supporting the hypothesis that those heads are functionally important.

Most closely related to our work are differentiable masking and soft-gating approaches that learn which attention heads to retain or suppress. In [47], the authors apply sparsity gating to identify subcircuits and use the fitted parameters as weighting values in convex combinations for activation patching. Similarly, [48] learns scaling constants for each attention head, but uses the fitted values to identify heads that are most suitable for fine-tuning. Thus, while methodologically similar, our work is unique in applying the gating parameters to identify task-sufficient causal sub-circuits.

Others [18, 40] have opted for hard, binary ablations using the Gumbel-softmax trick [49, 50], fitting gating probabilities rather than weighting parameters. Although these Gumbel-based approaches have been applied for causal circuit discovery in a similar spirit to our work, they suffer from a fundamental limitation that CHG does not. Specifically, while Gumbel-based gating methods also learn differentiable gates per head, they treat each head independently, effectively learning separate Gumbel–Bernoulli distributions for head inclusion. This factorized formulation models only marginal probabilities and cannot capture interdependencies between heads that jointly affect task performance. In contrast, CHG jointly optimizes all gating coefficients under the model's loss, capturing the full range of interactions and contingencies between the attention heads. Because CHG is highly scalable, it can be fit repeatedly across random seeds or subsets, effectively sampling from the space of sub-circuits without assuming independence between heads. This enables estimation of the underlying distribution over functional head configurations while preserving the joint statistical structure that factorized gating approaches discard.

# 3 Our Approach: Causal Head Gating

Causal head gating is based on three ideas: applying multiplicative gates to attention heads to evaluate their roles, using regularization to produce variation in the estimates of the gating parameters, and constructing a taxonomy based on that variation. We introduce these ideas in turn.

## 3.1 Applying gates to attention heads

For a transformer with $L$ layers and $H$ attention heads, we define a gating matrix $G \in [0, 1]^{L \times H}$, where $G_{\ell,h}$ scales the output of head $h$ in layer $\ell$, just before the output projection matrix $W_\ell^O$ (shown in red for an example head in Figure 1a). Given input hidden states $X \in \mathbb{R}^{\text{seq} \times d_{\text{model}}}$, each head computes:

$$A_{\ell,h} = \text{softmax}\left(\frac{XW_Q^{\ell,h}(XW_K^{\ell,h})^\top}{\sqrt{d_k}}\right), \quad V_{\ell,h} = XW_V^{\ell,h}, \quad Z_{\ell,h} = G_{\ell,h} \cdot (A_{\ell,h}V_{\ell,h})$$

where $W_Q^{\ell,h}, W_K^{\ell,h}, W_V^{\ell,h} \in \mathbb{R}^{d_{\text{model}} \times d_k}$ are learned projection matrices for queries, keys, and values.

The gating coefficient $G_{\ell,h}$ modulates the contribution of head $h$ by scaling its output $Z_{\ell,h}$ after attention is applied but before the heads are combined (see Figure 1a). The gated outputs are then

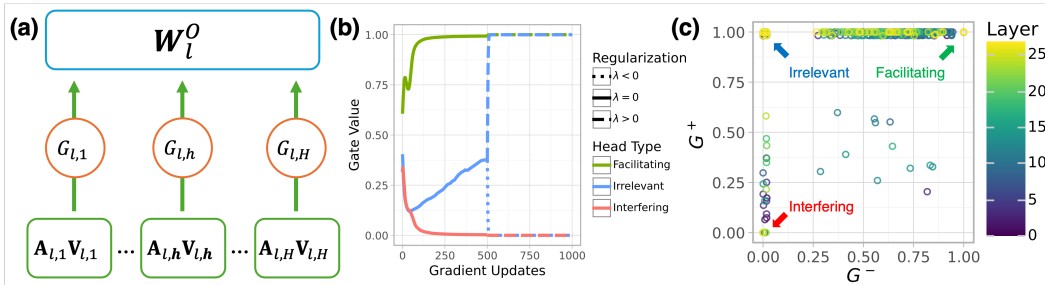

Figure 1: (a) Schematic of a single multihead attention block with CHG-determined gating attenuation (in red). (b) Gate fitting trajectories for three heads on L3.2-3BI with OpenMathInstruct2. When fitting with $\lambda < 0$ and $\lambda > 0$, $G^+$ and $G^-$ both stay near 1 for facilitating heads and near 0 for interfering heads, but bifurcate to 1 and 0 respectively for irrelevant heads. (c) Gate values after fitting.

Table 1: Causal taxonomy for head roles and corresponding gating patterns.

| Role | Description | $G^+$ | $G^-$ | Metric | Ablation Effect |
|------|-------------|-------|-------|--------|-----------------|
| Facilitating | Supports task performance | High | High | $G^-$ | Decreases task performance |
| Interfering | Interferes with task performance | Low | Low | $1 - G^+$ | Increases task performance |
| Irrelevant | Negligible impact on performance | High | Low | $G^+ \times (1 - G^-)$ | No effect on task performance |

concatenated and projected:

$$\text{Output}_\ell = \text{Concat}(Z_{\ell,1}, \ldots, Z_{\ell,H})W_\ell^O, \quad W_\ell^O \in \mathbb{R}^{Hd_k \times d_{\text{model}}}$$

We fit $G$ by freezing the parameters of the model $\mathcal{M}_\theta$ and minimizing the negative log-likelihood (NLL) on a next-token prediction task with a regularization term specified below.

## 3.2 Producing variation through regularization

We add a regularization term to the objective that introduces a small but consistent gradient—clipped to ensure NLL remains the dominant term—that nudges the gates for task-irrelevant heads toward 1 or 0 while leaving task-relevant ones relatively unaffected. The NLL optimizes towards improving task performance, and tunes the heads by either increasing the gating values for task-facilitating heads or decreasing the gating values for task-interfering heads. However, if a head does not affect task performance, i.e. is task-irrelevant, then the expected gradient from the NLL is 0, which confounds interpretation of task relevance when evaluating the tuned gating values: a gate $G_{l,h}$ may be close to 1 either because it is important for performing the task (causal), or because gating it has no effect (incidental). We address this limitation by introducing an $L_1$-regularization term in our objective function, with weight $\lambda$ that either nudges gates toward 1 for maximal density ($\lambda > 0$) or toward 0 for maximal sparsity ($\lambda < 0$):

$$\mathcal{L}(G; \mathcal{M}_\theta, \mathcal{D}, \lambda) = \underbrace{- \sum_{(x,y) \in \mathcal{D}} \log P(y \mid x; \mathcal{M}_\theta, G)}_{\text{Negative log-likelihood (NLL)}} - \lambda \underbrace{\sum_{i,j} \sigma^{-1}(G_{l,h})}_{\text{Regularization}} \tag{1}$$

where $\mathcal{M}_\theta$ is the model being analyzed, $y$ is the target text sequence for a given prompt $x$ in dataset $\mathcal{D}$, and $\sigma^{-1}$ is the clipped inverse-sigmoid function.

We fit $G$ twice: once with $\lambda > 0$ to encourage retention ($G^+$), and once with $\lambda < 0$ to encourage removal ($G^-$). To ensure that the heads are aligned across both optimizations, we first fit $G$ with $\lambda = 0$ to establish a shared initialization (see Figure 1), so that any differences between $G^+$ and $G^-$ reflect only the effect of the regularization and not divergent optimization paths.

## 3.3 Constructing a taxonomy of task relevance

The $G^+$ and $G^-$ matrices allow us to interpret the functional role of each head. To formalize this, we introduce a causal taxonomy (Table 1) in which each head is assigned one of three roles—*facilitating*, *interfering*, or *irrelevant*—based on its predicted impact on model performance under ablation. Facilitating heads positively contribute to performance, while ablating them degrades it. Conversely, interfering heads negatively contribute to performance, while ablating them improves it. Finally, irrelevant heads have negligible effect, with ablation leaving performance effectively unchanged.

We instantiate this taxonomy using the fitted CHG matrices $G^+$ and $G^-$, which reflect head behavior under opposing regularization pressures. Facilitation is measured by $G^-$: heads that remain active despite pressure to suppress are likely necessary for the task. Interference is measured by $1 - G^+$: heads that are suppressed even under encouragement to remain are likely harmful. Irrelevance is measured via $G^- \odot (1 - G^+)$, identifying heads that vary in gate values based on regularization.

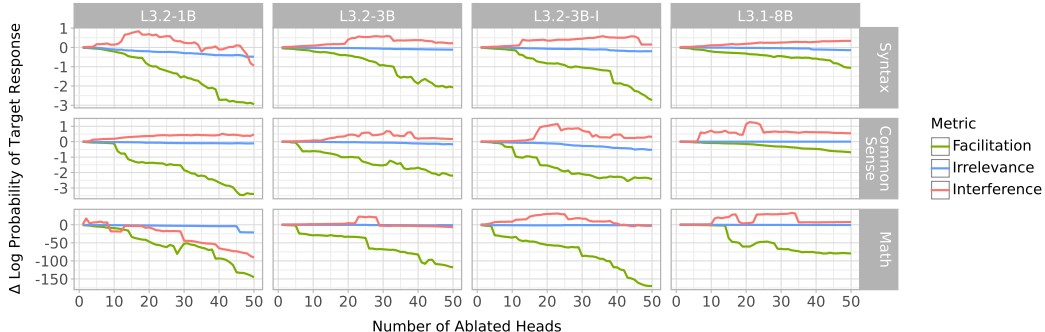

Figure 2: Difference in target log-probability when sequentially setting individual gates in $G^+$ to 1 and 0 in order of facilitation, irrelevance, and interference scores. The horizontal axis shows the number of heads ablated in descending score order. Positive values indicate task improvement, negative values indicate degradation, and values near zero indicate no effect. Note that not all heads in the top 50 necessarily have high absolute scores.

## 4 Experiments and analyses

### 4.1 Causal roles of attention heads

We begin by reporting experiments that evaluated the causal taxonomy presented in Table 1 across four variants of the Llama 3 LLM [3]: L3.1-8B, a pre-trained 8B-parameter model; L3.2-3B, a 3B-parameter model distilled from Llama-3.1-70B (not used in this paper); L3.2-3BI, an instruction-tuned version of Llama-3.2-3B; and L3.2-1B, a 1B-parameter model distilled from L3.1-8B. For each model, we fit CHG matrices on three task types performed over distinct datasets: mathematical reasoning from OpenMathInstruct2 [38], syntactic reasoning from the subset labeled "syntax" in BIG-Bench [51], and commonsense reasoning from CommonsenseQA [52]. We fit CHG matrices independently for each model-dataset pair across 10 random seeds.

We first test whether the causal scores align with the taxonomy's predictions about performance. Specifically, the taxonomy predicts that, when ablated, attention heads scoring highly on facilitation, irrelevance, or interference should decrease, leave unchanged, or increase the model's task performance, respectively. To test this, we sort heads in descending order by each causal metric and evaluate the model using the $G^+$ matrix while toggling each head to 0 or 1 in order of its score. While both $G^+$ and $G^-$ match the context in which scores were computed, we use $G^+$ as it retains more heads, providing a more interpretable baseline for ablation. We then compare the retained and ablated masks by the model's log-probability of the target sequence, expecting the resulting change in log-probability to follow the predicted pattern. As shown in Figure 2, these interventions match the predicted patterns: the difference in target log-probability is negative when progressively ablating facilitating heads, near 0 when ablating irrelevant heads, and positive when ablating interfering heads, up until the set of interfering heads is exhausted.

### 4.2 Distribution of causal roles

Having validated the causal scores using targeted ablations, we next analyze how they are distributed across models and tasks. Figure 3a shows that for each task, the distribution of head roles is highly consistent across all four model variants. This holds despite large differences in model size (1B to 8b) and training setup (pretraining, distillation, instruction tuning). We quantify these similarities by computing Pearson correlations between head scores across all model pairs for each task and causal metric, yielding 54 model pairs, all of which show high agreement with a minimum correlation of 94.92% and an average of 99.2%. Across tasks, however, we observe notable differences, with the math dataset standing out in particular. For syntax and commonsense reasoning, most heads are irrelevant—63.0% and 64.6% have irrelevance scores $\geq 0.5$, respectively—with only a sparse subset of facilitating heads (25.6% and 27.4% with facilitation scores $\geq 0.5$), suggesting that compact, redundant circuits are sufficient for these tasks. In contrast, mathematical reasoning activates a much larger fraction of facilitating heads: 52.6% have facilitation scores $\geq 0.5$, while only 39.0% are

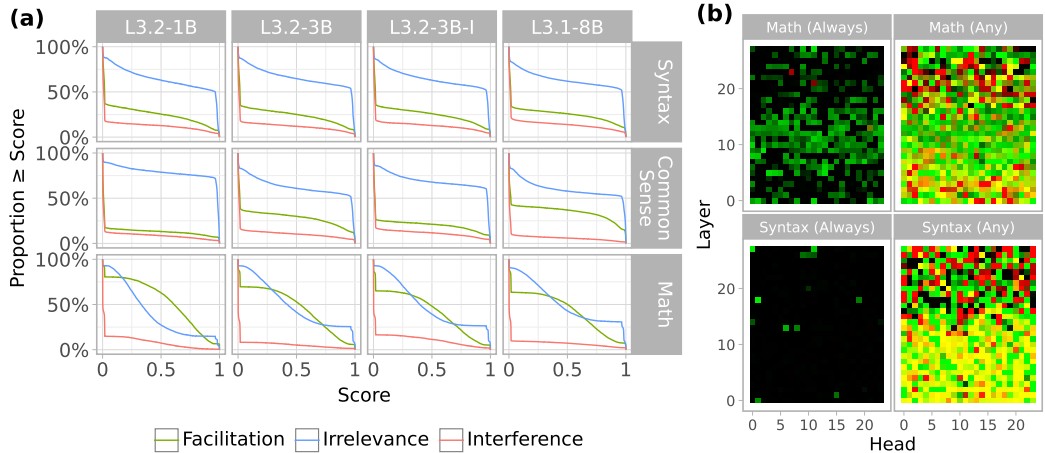

Figure 3: CHG score distributions and consistency. (a) Empirical cumulative distribution of CHG scores across all attention heads, showing the proportion of heads with scores below a given threshold for facilitation, irrelevance, and interference. (b) Aggregated CHG scores on L3.2-3BI, where red and green color channels represent interference $(1 - G^+)$ and facilitation $(G^-)$, respectively. Colors are combined using RGB rules: black indicates irrelevance (low in both), and yellow indicates both facilitation and interference (high in both). *Always* aggregates using the minimum across seeds (highlighting consistent effects); *Any* uses the maximum (highlighting any effect across seeds).

Table 2: Percent of heads with facilitation (F) or interference (N) scores $\geq 0.5$ across all seeds (always) or in at least one seed (any).

| Task | Agg. | L3.2-1B | | L3.2-3B | | L3.2-3BI | | L3.1-8B | |
|---|---|---|---|---|---|---|---|---|---|
| | | F | N | F | N | F | N | F | N |
| Syntax | Always | 1.2 | 0.2 | 1.5 | 0.1 | 0.7 | 0.0 | 1.4 | 0.0 |
| | Any | 72.1 | 57.2 | 67.9 | 51.3 | 72.8 | 56.1 | 68.5 | 59.2 |
| Common Sense | Always | 3.9 | 0.0 | 4.5 | 0.0 | 3.0 | 0.0 | 18.7 | 0.6 |
| | Any | 56.6 | 41.0 | 75.4 | 52.4 | 68.2 | 55.7 | 60.3 | 22.2 |
| Math | Always | 38.3 | 0.4 | 24.6 | 1.3 | 18.3 | 0.1 | 25.3 | 1.0 |
| | Any | 81.1 | 26.0 | 75.1 | 13.8 | 74.4 | 47.2 | 75.0 | 21.2 |

irrelevant, likely reflecting the task's higher complexity and need for broader sub-circuitry to support multi-step, latent computations.

It is also worth noting that, across all tasks, 84.0% of heads are marked as facilitating or interfering (score $\geq 0.5$) in at least one seed, yet only a small fraction are consistently facilitating or interfering across all seeds (Figure 3b). In syntax and commonsense tasks, most models have fewer than 5% of heads that are always facilitating and virtually none that are always interfering (Table 2). In contrast, math reveals more rigid and consistent circuitry, with up to 38.3% of heads consistently facilitating and 1.3% consistently interfering. These patterns suggest that individual attention heads may not have modular, context-independent roles, but instead participate in a flexible ensemble of overlapping sub-circuits, in which their function depends on the configuration of others [53].

## 4.3 Comparison with causal mediation analysis

CMA, like CHG, aims to identify attention heads that facilitate task execution, though it does so in a more hypothesis-driven manner. Framed in signal detection terms, CMA and CHG are complementary. CMA exhibits high precision but relatively low sensitivity: while many facilitating heads may go undetected (false negatives), those it does identify are reliably task-relevant (few false positives). Conversely, CHG is biased toward sensitivity over precision. This suggests that heads identified by CMA should also be identified (as showing strong facilitation) under CHG. We test this by comparing

CHG to the results of two former studies using CMA, replicating their methods to identify attention heads with specific computations: heads that encode task information in function vectors [12] and heads that perform symbolic reasoning [13].

For function vectors, we use the six in-context learning tasks used in [12]: 'antonym', 'capitalize', 'country-capital', 'English-French', 'present-past', and 'singular-plural'. Each prompt is presented in an in-context learning (ICL) [46] format consisting of 10 input-output examples using a "Q: X\n A: Y" template, followed by a query to be answered. To perform CMA, we corrupt the prompt by randomly shuffling example outputs to induce mismatched pairs, then patch individual head outputs with clean activations to identify which heads recover performance—interpreting high recovery as evidence of causal mediation.

We apply a similar logic to symbolic reasoning tasks from [13], where the goal is to generalize abstract identity rules such as ABA ("flowˆStartedˆflow") or ABB ("flowˆStartedˆStarted"). We deploy the same CMA procedure used in [13] to identify the three-stage symbolic processing mechanism that was reported: (1) *symbol abstraction* heads that abstract symbols ("A" or "B") away from the actual tokens in the in-context examples; (2) *symbolic induction* heads that operate over the abstracted symbols to induce the symbol for the missing token in the query; (3) *retrieval* heads that retrieve the actual token based on the induced symbol to complete the query. To screen heads of each type, we construct prompt pairs in which either the same token is assigned to different symbols ("A" or "B") or tokens are swapped while preserving the same rule, and patch activations at certain token positions between them. Attention heads that steer model behavior towards specific hypotheses about the three head types after patching (either converting the abstract rule or altering the actual token) are labeled as mediating. We conduct all experiments on the Llama-3.2-3B-Instruct model.

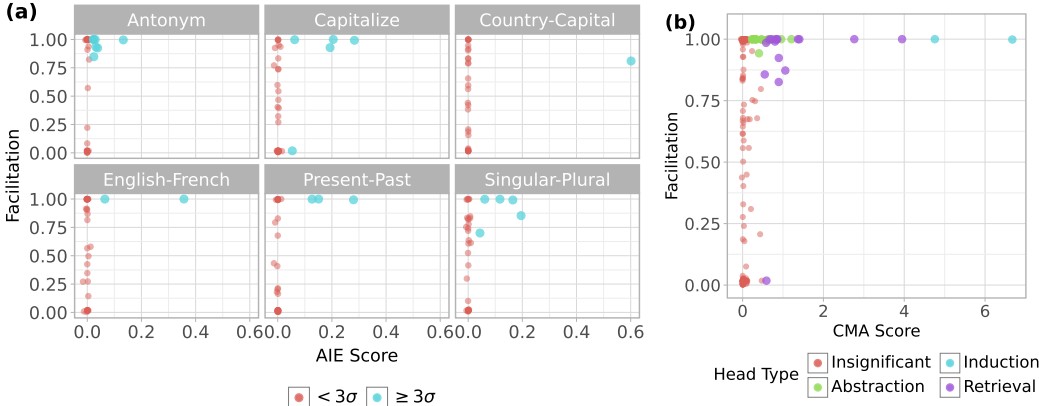

Figure 4: Task-facilitation scores versus (a) average indirect effect for function vector tasks and (b) CMA scores for symbolic reasoning tasks, showing significant heads by type (*abstraction*, *induction*, *retrieval*) and using the maximum CMA score across types for insignificant heads.

As predicted, CMA-identified heads tend to exhibit high facilitation scores under CHG in both domains (Figure 4). To quantify this, we compare the CHG facilitation scores of CMA-identified heads—those with three standard deviations above the mean in function vector tasks or with statistical significance in ABA/ABB tasks [13]—to the remaining ones. Since facilitation and irrelevance depend on the specific sufficient circuit identified by CHG, a head may appear irrelevant in one run but facilitating in another if multiple circuits exist. To account for this, we fit 10 CHG masks per function vector task and 20 per ABA/ABB task, and compute each head's maximum facilitation score across runs—capturing whether it participates in any sufficient circuit. We find significantly greater facilitation among mediating heads in both the function vector tasks ($t(23.05) = 8.52, p < 10^{-8}$) and the ABA/ABB tasks ($t(53.77) = 11.18, p < 10^{-15}$), supporting the relationship between CMA and CHG-identified task relevance.

## 4.4 Contrastive Causal Head Gating

The results above indicate that CHG effectively distinguishes among facilitating, irrelevant, and interfering attention heads. However, as an exploratory method, it lacks the granularity to charac-

terize the specific functions of these subnetworks. For instance, consider the 'antonym' task from Section 4.3, presented in an in-context learning (ICL) format with 10 examples and a single-word response, as defined in [12]. To perform this task successfully, the model must not only generate the appropriate antonym of a given word, but also infer the task itself from the 10 input-output pairs in the prompt. Thus, a minimal circuit of task-facilitating heads will contain both those involved in task inference and those involved in antonym production, and CHG cannot distinguish between the two. This becomes more pronounced as task complexity increases, as in the OpenMathInstruct2 dataset, where the minimal circuit must jointly support diverse sub-tasks, including English comprehension, mathematical reasoning, chain-of-thought processing, and LaTeX generation.

To address this, we introduce a simple extension of CHG that not only identifies facilitating heads for a given task but also isolates the sub-circuit responsible for a particular sub-task. We generate parallel variants of the same task that share all features except for a controlled difference in the required operation, allowing us to isolate the corresponding sub-circuits. In doing so, we take a step toward a hypothesis-driven approach, decomposing the task into sub-steps while remaining agnostic to the mechanistic implementations For example, the antonym task can be constructed as an ICL task using the default format from [12], or as an instruction-following task where the model is presented with the task description "Given an input word, generate the word with opposite meaning". By comparing the resulting attention circuits, we can disentangle components responsible for task inference from those involved in antonym generation.

Furthermore, rather than simply applying CHG to each version and directly comparing the results, we propose a combined approach that fits a single mask with a joint objective to forget one variant of the task while retaining the other, so that the resulting gate matrix suppresses heads uniquely necessary for one variant but dispensable for the other:

$$\mathcal{L}(G; \mathcal{M}_\theta, \lambda) = \sum_{(x_R, y_R, x_F, y_F)} \log P(y_F \mid x_F) - \log P(y_R \mid x_R) - \lambda \sum_{i,j} \sigma^{-1}(G_{l,h}) \quad (2)$$

where $\log P(y \mid x)$ denotes the log-probability of target sequence $y$ given prompt $x$ under model $\mathcal{M}_\theta$ with gating matrix $G$, the sum ranges over matched tuples $(x_R, y_R, x_F, y_F)$ of the retention and forget variants that differ only in task formulation, and $\lambda > 0$. To stabilize the gradient, we clip the inverse-sigmoid as in Eq. 1 as well as the difference in log-probability.

We evaluate this method using the six function vector tasks from Section 4.3, leveraging the natural language task descriptions provided in [12] to construct instruction-based variants. For each problem, we replace the 10-shot word-pair examples with a prompt containing the task instruction and a single example. We then fit the *contrastive* causal head gating (CCHG) mask to forget the ICL variant of five tasks while retaining the instruction-based format, holding out the sixth task for evaluation. If task inference from examples, instruction-following, and task execution are indeed mediated by separable circuits, this analysis should disable example-based generalization while preserving instruction-based performance. We perform our experiments in both directions (forgetting ICL while retaining instruction-following, and vice versa), using each of the six tasks as the held-out evaluation task. All experiments were conducted on the LLaMA-3.2-3B-Instruct model.

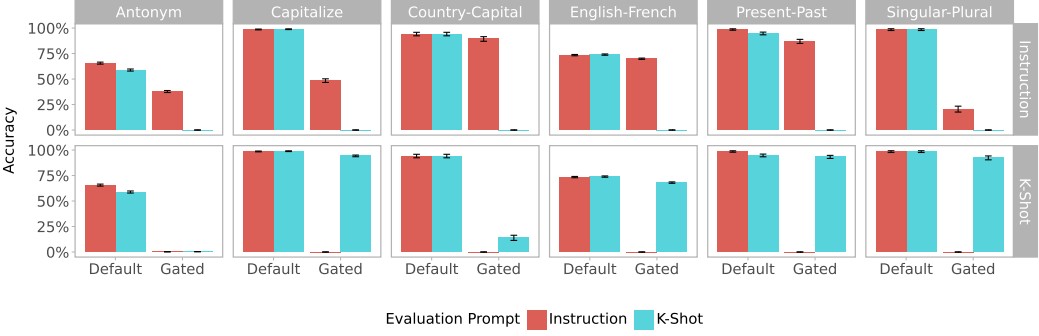

Figure 5: Task accuracy under CCHG. Columns indicate held-out evaluation tasks and rows indicate the retained prompt format. Bar color shows the evaluation prompt format. "Default" and "gated" indicate whether CCHG is applied during evaluation. Error bars indicate 95% CI.

As shown in Figure 5, the CCHG masks generalize to the held-out task. When the model is induced to forget task inference from ICL examples across five tasks, its target task accuracy drops to zero on the ICL variant of the held-out task while in most cases remaining well above zero—and often close to the unablated baseline—on the instruction-based variant. A similar pattern emerges when forgetting is applied using the instruction-based format: performance collapses on instruction prompts while generally remaining intact for example-based ones.

Interestingly, while degradation is often small for the retained prompt format, this pattern is not consistent across all tasks. For example, when the gating matrix is fitted to retain ICL and forget instruction-following, the 'singular-plural' task shows only a small drop in ICL accuracy ($98\% \rightarrow 92\%$) but a complete failure on instruction prompts ($98\% \rightarrow 0\%$). When this setup is reversed—fitted to retain instruction-following and forget ICL—accuracy on ICL drops from 98% to 0%, while instruction accuracy drops more modestly ($98\% \rightarrow 21\%$). Across the 6 tasks, 3 ('country-capital', 'English-French', 'present-past') remain robust as held-out tasks under instruction prompts, and 4 ('capitalize', 'English-French', 'present-past', 'singular-plural') do so under ICL prompts.

Thus, our results indicate that the circuits for instruction following and ICL may be separable at the head level. However, this separability also depends on the task, suggesting that task execution circuits may share heads with those used for task understanding and representation.

# 5    Discussion

In this work, we introduced Causal Head Gating (CHG), a flexible and scalable method for identifying causally relevant attention heads in large language models. CHG assigns each head a graded score for facilitation, interference, or irrelevance based on its effect on task performance, going beyond correlational or observational analyses. These scores predict performance changes under targeted ablations, confirming that facilitation, interference, and irrelevance scores capture causal impact. Crucially, it does so using next-token prediction alone, thereby avoiding reliance on labeled data or handcrafted prompts, making it broadly and easily applicable. Moreover, CHG requires no finetuning or auxiliary decoder model, and introduces only one parameter per head, allowing it to run in minutes even on billion-scale models. To validate our method, we demonstrated that existing works within the mechanistic interpretability literature successfully corroborate our findings using, and that the ICL and instruction-following circuits revealed using contrastive CHG successfully generalize across tasks.

Interestingly, across the range of models and tasks we investigated, we observed that attention heads form task-sufficient sub-circuits with low overlap. Moreover, a single head may vary in its relevance across multiple runs depending on which others are active, reflecting the distributed and context-dependent nature of computation in LLMs, and in rare cases, a head may even receive low $G^+$ but high $G^-$ scores within the same run. We hypothesize that this variability reflects an interaction-dependent landscape in which causal roles shift with circuit configuration. While these complexities may appear messy, we view them as a strength of CHG, revealing the redundancy and interdependence that underlie emergent model behavior. Because CHG is highly scalable, it can be repeatedly applied to estimate distributions over gating values, providing a bootstrapped view of redundant and contingent sub-circuits with greater fidelity to the model's underlying dependency structure.

While CHG provides a lightweight and scalable approach for exploratory analysis, requiring only a dataset and no model finetuning or supervision, it is not designed to reveal the precise computations performed by individual heads. Instead, CHG offers a complementary first-pass diagnostic tool that identifies candidate heads or sub-circuits with consistent causal influence, guiding where more granular, hypothesis-driven methods such as causal mediation or activation patching can be applied. In this way, CHG provides a practical entry point into large-scale causal interpretability, mapping functional dependencies that subsequent analyses can examine in greater detail.

We hope that our work encourages further exploration of causal structure in language models as a foundation for more mechanistic understanding. Future work may build on these tools to develop circuit-level explanations of how models implement complex behaviors.

## Acknowledgments and Disclosure of Funding

We thank Declan Campbell and Alexander Ku for helpful discussions, and Legasse Remon for assistance with dataset organization.

Jonathan Cohen was supported by the Vannevar Bush Faculty Fellowship, sponsored by the Office of Naval Research.

The authors declare no competing interests.

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

# 6 Technical Appendices and Supplementary Material

## 6.1 Datasets

For each dataset, we split the full set into three partitions: an example set, a training set, and a validation set. Example problems were selected from the top $K$ shortest prompt-solution sequences after tokenization. One example was randomly drawn from the example set to be included in each training/validation prompt to help align model responses with the task format. For multiple-choice datasets, answer options were randomly shuffled and labeled with capital letters (A, B, C, ...), and the target answer was the correct letter.

**OpenMathInstruct2** We used the `OpenMathInstruct-2_train_1M` subset. We filtered out problems marked as having no solution, removed duplicate prompts (even if their solutions differed), and retained the 55,050 shortest prompt-solution pairs by total tokenized length. From this, we selected 50 examples, 50,000 training problems, and 5,000 validation problems. Each prompt began with the instruction: "For each problem, explain your reasoning step by step and use LaTeX for all mathematical expressions. Indicate your final answer using \boxed{...}."

**CommonsenseQA** We selected 10 problems for the example set, then split the remaining data into a 90% / 10% training/validation split.

**BIG-Bench syntax** We included all tasks labeled 'syntax' in BIG-Bench: 'linguistic mappings', 'tense', and 'subject-verb-agreement'. The 'linguistic mappings' category consisted of five subtasks, each with its own instruction:

- **Past tense**: "Convert the verb to its past tense form."
- **Plural**: "Convert the noun to its plural form."
- **Pronoun replacement**: "Replace the repeated name with the correct pronoun."
- **Question formation**: "Convert the statement into a yes/no question."
- **Sentence negation**: "Convert the statement into a negative sentence."

The 'tense' task used the instruction: "Modify the tense of a given sentence."

The 'subject-verb-agreement' task used the instruction: "Choose the grammatically correct verb form that agrees with the subject of the sentence."

Each task or subtask was treated independently for splitting and prompt generation. We allocated 10 examples per subtask, with a 90% / 10% split over the remainder into training and validation. Example problems used in prompts were always drawn from the same subtask as the target problem.

**Function vector tasks** We included six tasks: 'antonym', 'capitalize', 'country-capital', 'english-french', 'present-past', and 'singular-plural'. Each task was used in two formats: 10-shot in-context learning (ICL) prompts with 10 input-output pairs, and instruction-based prompts using task descriptions from [12]:

- **Antonym**: "Given an input word, generate the word with opposite meaning."
- **Capitalize**: "Given an input word, generate the same word with a capital first letter."
- **Country-Capital**: "Given a country name, generate the capital city."
- **English-French**: "Given an English word, generate the French translation of the word."
- **Present-Past**: "Given a verb in the present tense, generate the verb's simple past inflection."
- **Singular-Plural**: "Given a singular noun, generate its plural inflection."

We allocated 10 examples per task, and split the remaining data into 90% training and 10% validation. Example problems used in prompts matched the format (ICL or instruction) of the task being evaluated.

**Symbolic reasoning (ABA/ABB)** : We procedurally generated symbolic reasoning prompts following the A^B^A and A^B^B templates from [13]. using 4 in-context examples per prompt. Each prompt was generated by selecting 10 random tokens—8 assigned to the 4 examples and 2 used in the query. We used individual tokens rather than full words, since multi-token words often behave similarly: once the first token is generated, the model tends to complete the rest automatically, reducing the task to token-level pattern recognition.

## 6.2 Training details

**Causal head gating**  For each model and task, we first fit a CHG matrix $G$ with $\lambda = 0$ for 500 gradient updates with a batch size of 64 samples. $G$ was initialized with random values sampled uniformly between 0 and 1. We used the Adam optimizer [54] for optimization using an initial learning rate of 0.1 with a linear decay that terminates with a learning rate of 0.01. After fitting $G$ with $\lambda = 0$, we fit $G^+$ and $G^-$ using $G$ as the initial conditions and $\lambda = \pm 0.1$ for 500 gradient updates with an initial learning rate of 0.5 and a terminal learning rate of 0.1. We clipped the regularization term at $\pm 4$.

**Contrastive causal head gating**  For each model and task pair, we fit a CCHG matrix $G$ with $\lambda = -0.1$, clipping the regularization term at 4 and the log-probability difference at 5. We fitted $G$ over 500 gradient updates with a batch size of 64 using the Adam optimizer with an initial learning rate of 0.1 with a linear decay that terminates with a learning rate of 0.01.

## 6.3 Hardware and compute

For all our experiments, we used 128 GB of CPU RAM and a single Nvidia H100 GPU at a time. Each run of CHG (1,500 gradient updates) took between 15 minutes and 1 hour, depending on the model and dataset. Each run of CCHG (500 gradient updates) took approximately 5 minutes. We estimate that all experiments reported in this paper can be completed in under 100 GPU hours. Preliminary or failed experiments required negligible additional compute and are not included in the total.

