# OpenReview forum: "Causal Head Gating: A Framework for Interpreting Roles of Attention Heads in Transformers"
_NeurIPS.cc/2025/Conference — NeurIPS 2025 poster_

### Official Review · Reviewer_ymfF · 2025-06-23

**Clarity:** 4
**Significance:** 3
**Originality:** 2
**Rating:** 5
**Confidence:** 4

**Summary:**

This paper introduces causal head gating (CHG), a method that determines the relevance of individual attention heads for performing a given task. The method trains two masks using standard cross-entropy loss on a task dataset w/ different regularization penalties, and based on the mask values, CHG labels an attention head with one of three categories: facilitating, interfering, or irrelevant. Using CHG, the authors identify attention heads important for three different kinds of tasks: math, syntax, and commonsense reasoning. They find that the role and importance of attention heads can vary based on other heads and the context. They also find that the heads identified by CHG causally affect generation, and that CHG scores correlate strongly with causal methods of attention head localization.

**Questions:**

As I stated in the weaknesses section, my main concern is related to the contextualization of the method - both providing baselines and comparisons to previous work in terms of design choices. This leads to the following questions:

1. How does your method compare to previous methods such as Voita et al. [1], Li et al. [2] that also learn gate coefficients for individual attention heads? Does your method find more/less heads? Does CHG improve performance under intervention more than previous work? Does CHG take the same amount of time to converge?, etc. (see weaknesses for more detail)

2. When designing CHG, what other design choices did you try that perhaps did not work as well? For example, how did you decide you needed to differentiate heads using two runs w/ a flipped learning objective?

**Ethical Concerns:**

["NO or VERY MINOR ethics concerns only"]

**Final Justification:**

During the rebuttal phase, the authors addressed my concerns about the framing/discussion of related work and how the proposed method differs in theory. I also found helpful their clarification about the consistency of CHG over multiple seeds. Incorporating this information into the main paper will help clarify the framing and contributions of the method for future readers.

I still maintain that additional empirical validation of the method and comparison to previous sparsity methods would strengthen the results even more (hence not a higher score), but the contributions are sufficient in my mind to warrant acceptance.

**Limitations:**

Yes, limitations are thoughtfully discussed.

**Quality:**

3

**Strengths And Weaknesses:**

**Strengths:**
- The paper is written clearly and was fairly easy to follow throughout.
- The method does not rely on paired data, which is a nice complement to other localization methods like causal mediation, which often relies on counterfactual pairs. The result showing that CHG scores correlated with causal-based scores is further evidence that these methods might complement each other well for future analyses. The flexibility of the method (that you can construct a contrastive version) is also nice.
- The claims in the paper generally have evidence to back them up, though some results are stronger than others. (e.g. results comparing ICL and instructions are a bit muddled, with some tasks exhibiting certain behavior, and others not having the same property)

_______
**Weaknesses:**

My main concern is a lack of comparison with previous work, despite many in-text citations. The authors are aware of related work (including very recent papers), and I would say this a strength of the paper. However, there is a lack of comparison to previous work in two places that I think would help significantly enhance the context of the paper (some places do compare, like the comparison with CMA section, and this is a nice contribution).

- First is in the related work section itself. More effort to differentiate your work from previous related work in the related work section would be helpful. In each paragraph of the related work, there is a lack of discussion about why your work/method is related to or different from the cited work. For example, probing (representational decoders) is discussed for a whole paragraph, but there is no probing done in this paper. While this section is a nice review, contextualizing your work here would improve the paper. The other related work sections are similar.

- The second place I think more comparison to related work would help is in the definition and evaluation of the CHG method. The CHG method is very similar to other previous methods that also learn gate coefficients per attention head. (e.g., Voita et al. [1],  Li et al [2], etc.). While these other papers did so in the context of pruning, it’s feasible to adapt them to the settings you’re evaluating here. I would recommend comparing and contrasting your method with these previous related methods. This would help provide some more justification for the choices you made while designing your CHG method, and give some more context as to how it compares to previous attempts in sparse optimization. As far as I could tell, the main difference of your method is the use of L1 regularization to induce sparsity instead of L0 penalty + annealing the gate values via the gumbel-softmax trick? The insight that learned gate values may end up close to 1 either because of task-importance or incidentally is an interesting one, and needing to run 2x to remove incidental importance seems to overcome that. It's likely this is a general problem of sparse optimization but it’s unclear from the current writing & experiments whether previous gate-learning methods are able to overcome this or not. This is where comparison might help show the advantage of CHG.

- While the CHG mask is jointly learned and implies some sort of interaction it does not detail the kinds of interactions that the attention heads selected may have. This is one limitation that is not discussed explicitly, but may fall under the idea that this method does not identify the specific functional role of any attention heads, only whether they facilitate or inhibit performance of the dataset task.
___
**Other:**
- It was unclear in Figure 1 which task, dataset & model are used for the plots.
- There are also a few related works (Davies et al. [3], Cao et al. [4]) that learn differentiable masks over components that were not cited but may be of interest to the authors. Several times the authors point to the idea that attention heads are dependent on interactions with other heads. There are a few works that describe and study such interactions between attention heads (Elhage et al. [5] or Merullo et al. [6]), though the second one is not cited and may be of interest.

- A minor typos on Line 217, there's an extra “that”
___

[1] Voita et al. [Analyzing Multi-Head Self-Attention: Specialized Heads Do the Heavy Lifting, the Rest Can Be Pruned](https://aclanthology.org/P19-1580.pdf)

[2] Li et al [Differentiable Subset Pruning of Transformer Heads](https://aclanthology.org/2021.tacl-1.86.pdf)

[3] Davies et al.  [Discovering Variable Binding Circuitry with Desiderata](https://arxiv.org/pdf/2307.03637)

[4] Cao, et al. [Low-complexity probing via finding subnetworks](https://aclanthology.org/2021.naacl-main.74.pdf)

[5] Elhage et al. [A Mathematical Framework For Transformer Circuits](https://transformer-circuits.pub/2021/framework/index.html)

[6] Merullo et al [Talking Heads: Understanding Inter-Layer Communication in Transformer Language Models](https://proceedings.neurips.cc/paper_files/paper/2024/file/70e5444e5f331f7f5431f302110b97af-Paper-Conference.pdf)

---

> ### Author Rebuttal · Authors · 2025-07-31
>
> We thank the reviewer for insightful comments. Given the rebuttal rules that prohibit resubmission of any form, we will do our best to answer the reviewer concerns here and hope that our responses would help appeal for a higher reviewer score.
>
> 1.	Re: comparison to Gumbel-softmax based methods (Voita et al. and Li et al.) – We appreciate the reviewer raising this important point. While we cannot provide new experimental data during this rebuttal period, we can provide a more theoretically grounded distinction between CHG and Gumbel-softmax based methods.
> While both CHG and Gumbel-based methods learn a parameter per head, they differ fundamentally in how they handle dependencies between heads. Gumbel methods model each head independently, treating each gating parameter as defining a separate Bernoulli distribution. As a result, they learn marginal distributions over head inclusion, without accounting for how heads may functionally depend on each other. In an LLM with N heads, the optimization effectively learns N different distributions. However, since the gate coefficients are sampled simultaneously, they model marginal distributions that preclude modeling interdependencies between heads.
> To illustrate, imagine heads A and B both perform the same role. A good sparse solution might activate one and suppress the other, but not both. Gumbel methods would assign roughly P(A) = P(B) = 0.5, resulting in equal probability to all combinations—including ones that activate both or neither—because they don’t capture the interaction.
> CHG, by contrast, jointly optimizes all gating coefficients under the model's loss. This means CHG naturally captures interdependencies: the value for head A can depend on the value for head B, and vice versa. In other words, CHG effectively samples from a coupled energy landscape, capturing dependencies that factorized gating methods cannot represent.
> That said, it is possible to modify Gumbel strategies to account for dependencies (e.g., by sampling sequentially), but this adds complexity and deviates from standard approaches like those in Voita et al. In practice, our results show that CHG often assigns different roles to the same head across seeds, which further suggests the presence of interdependencies that marginal models like Gumbel might miss.
> While we leave full empirical comparison to future work, we believe CHG offers a meaningful theoretical advantage by modeling head interactions directly. We hope this distinction helps clarify the unique value of our method.
> 2.	Re: related work – Thank you for your comment on the related work, as well as some of the suggested citations. Our aim here was to compare with other interpretability tools used in the literature more broadly and agree in retrospect that there could have been a more direct comparison with the most adjacent methods, especially in the theoretical motivation for CHG that contrasts it with the Gumbel based methods. If permitted, we will incorporate this discussion into the final publication.
> 3.	Re: Figure 1 - These plots were generated from applying CHG to the Llama 3.2 3B-Instruct model on the OpenMathInstruct2 dataset.
> 4.	Re: design choices - Our decision to use two runs was theoretically, rather than empirically, motivated by the goal of distinguishing between task facilitating, irrelevant, and interfering heads. A single optimization could distinguish one from the other two (e.g. facilitating from irrelevant and interfering in the G- optimization), but not all three. The only other design choices we explored were hyperparameters (e.g. the lambda values) which did not significantly impact the results (within reasonable values).
>
> We appreciate the generally positive scores on quality, clarity, significance, and originality. If our responses to the reviewer concerns are satisfactory, we hope that you would reflect this in a higher overall rating for our work.

---

> ### Comment · Reviewer_ymfF · 2025-08-04
>
> Thank you for your response. The clarification of your thoughts on the difference between CHG and similar methods is helpful and would be a nice addition as a discussion in the related work.
>
> > "our results show that CHG often assigns different roles to the same head across seeds"
>
> Thanks for mentioning this in the rebuttal. I wanted to get some additional clarity about this point, as this inconsistency is one of the major limitations of the proposed CHG method. As I think about it some more, it seems like this suggests that CHG in fact has a hard time modeling interdependence of heads, though this is claimed as one of the theoretical strengths of the method ("CHG naturally captures interdependencies"). Can you clarify CHG's inconsistency of role assignments across seeds a bit more? For example, how many seeds you would need to run to be sure you capture the true role/effect? (e.g., how many seeds are used for the results in 4.3?) Does this limit the reliability of CHG in some ways? Or maybe it points to the fact that localization is much more messy than previously thought due to redundancy and interdependence.
>
> Based on the rebuttal, I appreciate the comments related to related work and feel like my questions will be/have been addressed there. However, I still have some reservations about the empirical validation of the method (comparison with similar previous methods), and am looking forward to a discussion of the above point. Overall, while the current results are promising, I'm inclined to keep my score for now until further discussion.

---

> > ### Author Response · Authors · 2025-08-05
> >
> > Thank you for your response and the follow up question. The concern about consistency across runs was also raised by Reviewer foUj, who found our response satisfactory enough to provide a higher reviewer score. We hope you find our response equally convincing.
> >
> > 1. Reproducibility across seeds: We emphasize that consistency across seeds is not the objective. Variability in gating values reflects distributional variance, not irreproducibility. As we illustrated with the example in our previous response regarding heads A and B that perform similar roles, so that they are redundant for the purposes of a given task. We would like to not only find a configuration where A is active and B is suppressed, but also the configuration where B is active and A is suppressed. Due to CHG’s high scalability, one can efficiently run multiple seeds to capture a wide distribution of redundant and interdependent configurations, preventing the practitioner from being misled into treating any single mask as the ‘true’ task subcircuit.
> > 2. Recommended number of seeds: CHG is best viewed as a bootstrapping method for estimating a distribution over gating values, rather than identifying a single optimal configuration. Thus, the number of CHG fits depends on the model, task, and head. Some heads may have sufficiently modular or task-necessary functions that they are invariant to the initializations, whereas others have higher interactions with other heads. Either result may be interesting, depending on the practitioner’s goals. If the objective is to capture the distribution of causal scores for each head irrespective of their inter-head interactions, 20–30 fits is likely sufficient, as this is equivalent to estimating distributions bounded between 0 and 1. For broader analyses using min and max values as used in this work, 10 is likely sufficient.
> > 3. Reliability and cross-validation – Our cross-validation results with two existing CMA-based papers in Section 4.3 as well as the cross-task generalization properties shown in Section 4.4 suggest that CHG is able to identify heads that have stable, convergent roles. In other words, while not every head in an LLM may be modular, our experiments demonstrate that CHG is able to reliably identify heads (with just 10 runs in our experiments) that are causally relevant, modular or not.
> > 4. Future directions: We wholly resonate with your hypothesis: "maybe it points to the fact that localization is much more messy than previously thought due to redundancy and interdependence." While the simpler and more straightforward results presented in this paper provide confidence in the validity of the approach, it is the messy complexities that raise more interesting questions about the emergent properties in LLMs and their underlying mechanisms. We hope that the reviewer shares our sentiment in appreciation towards possible future directions suggested by the unanswered questions in our work rather than simply as critical limitations.
> >
> > We hope this clarifies that CHG is both reliable and intentionally designed to capture distributed interdependencies, and we would greatly appreciate your consideration of this clarification in your final assessment.

---

> > > ### Comment · Reviewer_ymfF · 2025-08-05
> > >
> > > Thank you for the reply. I appreciate the explanation behind how you view your method, which was not clear upon reading the paper. I look forward to the incorporation of these thoughts in the final draft (as well as the previously discussed changes to the related work) and have updated my score, as all of my questions have been answered during the rebuttal. I still think comparing CHG's findings to previous sparsity methods would strengthen the empirical results in the paper, but the contributions are sufficient in my mind to warrant acceptance.

---

> > > > ### Author Response · Authors · 2025-08-08
> > > >
> > > > Thank you for the thoughtful discussion and the updated score. If accepted, we will do our best to incorporate any clarifications and suggestions raised during this period into the final version.

---

### Official Review · Reviewer_PMP1 · 2025-06-25

**Clarity:** 3
**Significance:** 2
**Originality:** 3
**Rating:** 3
**Confidence:** 3

**Summary:**

The authors present Causal Head Gating (CHG), a scalable method for interpreting the functional roles of attention heads in transformer models. The method involves learning a matrix of soft gates, one for each head, by optimizing the standard next-token prediction loss on a given task dataset, augmented with a special regularization term. They also introduce a contrastive variant to isolate circuits for specific task components.

**Questions:**

1. How should one interpret the "causal role" of a head that is "facilitating" in some runs and "irrelevant" in others?
2. The gating matrix G is optimized for a specific task dataset D. Have the authors considered applying a G matrix trained on one task (e.g., mathematical reasoning) to a completely different task (e.g., commonsense QA)? This would provide evidence about the degree to which the identified "circuits" are task-specific versus representing more general computational primitives.
3. Regarding the contrastive CHG experiment, the gating matrix G itself is quite expressive. How can we be certain that the method is truly discovering a pre-existing functional separation between ICL and instruction-following circuits, rather than G learning to create or simulate this separation to satisfy the contrastive objective?

Typos:
* Line 3: over heads -> overheads
* Line 17: state-of-the art -> state-of-the-art
* Line 46: interpretablity -> interpretability
* Line 217: This suggests that that -> This suggests that

**Ethical Concerns:**

["NO or VERY MINOR ethics concerns only"]

**Final Justification:**

After discussions, my main concerns remain. The paper's claims about "causality" are overstated, as the method deals with predictive correlations on a fixed dataset, and the key experiments (Fig. 5) fail to provide sufficient evidence for the invariance or generalizability of its findings.

**Limitations:**

The instability and ambiguity of the "causal roles" assigned by CHG as mentioned in weaknesses.

**Quality:**

2

**Strengths And Weaknesses:**

Strengths:
1. The proposed method, CHG, is highly scalable and does not require manually crafted prompt templates or external labels beyond what is used for next-token prediction.
2. The contrastive CHG extension is interesting, offering a way to start disentangling the circuitry for different sub-tasks.

Weaknesses:
1. I find the “causal” claims in the paper overstated and potentially misleading. When comparing their work with SAE and probing classifiers, they say “(SAE’s) insights remain correlational and still depend on post hoc labeling or interpretation, inheriting the same supervision bottleneck”. However, I don't see how CHG is more causal than SAE, given that the concept of causality in both is the intervenability of latent/transformed variables in a neural network considered as a causal graph. CHG is less causal than SAE in two senses: (i) CHG analyzes coarse, polysemantic units (heads) and gets unstable role assignments, whereas SAEs aim to find fine-grained, and potentially monosemantic units. (ii) while SAE aims to interpret/control the output (as the effect of the intervention), CHG only interpret/control which head contributes to the final loss (as the effect of the intervention) on a specific data distribution.
2. The paper's own results demonstrate that the assigned causal roles are not stable and vary across different random seeds. This suggests that the discovered roles are not intrinsic properties of the attention heads themselves, but rather artefacts of the CHG optimization process converging to one of many possible sufficient sub-circuits. This instability undermines the claim of “identifying” a “causal” attention head.

---

> ### Author Rebuttal · Authors · 2025-07-31
>
> We thank the reviewer for insightful comments. Given the rebuttal rules that prohibit resubmission of any form, we will do our best to answer the reviewer concerns here and hope that our responses would help appeal for a higher reviewer score.
>
> 1.	Re: CHG abstraction level – We agree that CHG operates at a coarser level than SAE. Our goal is not to match the granularity of representational methods but to work at the head level, which we view as a meaningful and tractable unit for analyzing model function. Attention heads offer useful levels of abstraction over individual representational units in the same way that brain regions may serve as useful abstractions over individual neurons. If the practitioner’s goal was to understand the role of individual representational units, then we are in agreement that SAE is likely to offer more useful insights than CHG. We present CHG not necessarily as a direct substitute for SAE, which is useful in its own right, but as a potentially complementary method that operates at a higher level of abstraction.
> 2.	Re: CHG and downstream control – We respectfully disagree with the claim that CHG does not affect model outputs. CHG gates control whether a given head is active, and the resulting change in the model’s behavior is directly measured through the loss. Since the loss aggregates effects on the model’s predictions, CHG does intervene on internal components and evaluates the causal impact at the output level—just as any intervention-based causal analysis should.
> 3.	Re: SAE and causality – The limitation we raise by discussing SAE in juxtaposition with representational decoders is that like decoders, SAEs do not necessarily discriminate between incidental values and causal values. Just because a representational unit in a network has a particular value does not guarantee that the model necessarily uses it to perform the task. For example, the problem “John has 3 red apples and eats one. The number of remaining red apples is…” would likely produce some representation of ‘redness’ due to the semantic content of the apples, despite color being irrelevant to the task. While an SAE, which aims to preserve all information, could yield monosemantic units that can be intervened on, this would require a priori knowledge from the user to know that ‘3’ is task-relevant while ‘red’ is not. This is not to say that SAE cannot be used to identify causal components; one certainly could with careful experimentation and appropriate controls. Our message here is that CHG is directly causal (via interventions), whereas SAE in its standard form is more observational (interpreting what’s already there).
> 4.	Re: CHG stability across seeds – We would like to clarify that variability in CHG coefficients across seeds is not instability but an intentional design choice. CHG is best viewed as a bootstrapped estimator over a distribution of sufficient subcircuits. Different fits reveal different but valid configurations of causal head usage, reflecting the redundancy and contingency in how transformer models distribute function. This distributional perspective provides insight into which heads are essential across configurations, versus which are involved only under certain inter-head dependencies.
> 5.	Re: causal role differences across seeds - We hypothesize that this form of variance across seeds would occur if heads are either redundant or contingent. First, consider two heads A and B that perform the same function, so that either is sufficient for the task and having both does not substantially improve performance. In a seed where the initial CHG coefficient of A is higher than B, CHG would likely identify A as facilitating and B as irrelevant (redundant), and the reverse in the opposite case. Next, again consider heads A and B where the function of head A is contingent on B, so that if B is suppressed, A has function X and Y if B is not. In this case, A may be considered facilitating if B is enabled but irrelevant if B is not, reflecting the changing roles. We hope to answer these differences in further research.
> 6.	Re: applying gates fitted on one dataset on a different dataset – We have observed that gating matrices fitted on BigBench Syntax, OpenMathInstruct2, or CommonsenseQA do not transfer well when applied to a different dataset. This is expected, as the three datasets target distinct capabilities within the model, and the learned gating patterns are correspondingly different. As seen in Figure 3a, even within the same model, the CHG score distributions vary substantially across datasets. The exception lies in the contrastive method in Section 4.4, where isolating heads responsible for instruction following or ICL allows for generalization to other tasks. This suggests that CHG captures task-specific subcircuits by default, but that contrastive CHG can help surface more generalizable, reusable components.
> 7.	Re: expressivity of G - This is a valid and important concern, which is why we validate the learned gating matrices by holding out entire tasks in the contrastive CHG experiments. If G simply overfit to the contrastive objective, we would not expect any meaningful transfer to a held-out task. However, as shown in the Present–Past experiment, ablating instruction-following or ICL heads—identified using five other tasks—can selectively impair the corresponding mode of operation without affecting the other. This suggests that the learned gates are isolating functionally meaningful circuits. That said, this effect is imperfect (see Figure 5), and we are actively investigating why some tasks permit clean separation at the head level while others do not. Our current hypothesis is that certain tasks share heads with instruction-following or ICL circuits, so ablating those heads inadvertently disrupts task performance.
>
> We appreciate the generally positive scores on quality, clarity, significance, and originality. If our responses to the reviewer concerns are satisfactory, we hope that you would reflect this in a higher overall rating for our work.

---

> > ### Comment · Reviewer_PMP1 · 2025-08-04
> >
> > Thanks for the reply and for explaining your view. I have read it, but I'm still not convinced about the paper's main claims for a couple of reasons.
> >
> > 1. You pointed out that SAEs might not distinguish between causal and incidental values. I think CHG has the same problem.
> > Imagine your dataset has a quirk where problems with the name "John" often have an even number answer. To get a low loss score, your method would have to learn this pattern. So, CHG would flag the heads that notice "John" as "facilitating". But that's clearly not a real cause-and-effect relationship; it's just a spurious correlation in the data. This shows that the method can be misled by spurious correlations present in the data.
> >
> > 2. A key criterion for a causal claim is invariance—the idea that a causal mechanism should remain stable across different contexts. Your own results show this isn't the case here.
> > You agree that the roles found for one task mostly don't transfer to other tasks. The main test for this was in your contrastive CCHG experiment (Figure 5), where you checked if the "circuits" worked on a new, held-out task. That test seems to have mostly failed, showing the findings don't generalize well.
> >
> > To sum up, I think the paper is making claims about causality that are too strong. The method is interesting for finding predictive patterns, but the evidence doesn't support that these patterns are causal. For these reasons, I will be keeping my score.

---

> > > ### Author Response · Authors · 2025-08-05
> > >
> > > Thank you for your response and your follow up points. We believe that the reviewer's concerns stem from certain misunderstandings about our work that we hope to clarify in this response.
> > >
> > > 1. **Spurious correlations:** We would like to clarify that the causality that we highlight in this work is not between tokens in the corpus, but between model components and the task. We agree with the reviewer that if the dataset is skewed to reflect spurious correlations, our method will not be able to disentangle these relations. However, our goal is not to disentangle correlations from causality in the data; it is in the model that we are interested in.
> > >
> > >     This is a critical distinction between CHG and SAE that is directly reflected in the loss function. SAE is trained to retain the existing information that is represented in the LLM already, spurious or not. If the model represents task irrelevant information, SAE will necessarily need to encode that to reconstruct the original vector. On the other hand, CHG's loss function is about the task, so if any model component during the task execution is irrelevant, CHG will be able to identify it as such. In other words, whereas SAE defers causal judgment to the practitioner (what in the representation is causally relevant and what is not), CHG incorporates that into the method itself.
> > >
> > >     Moreover, it is also worth noting that because CHG abstracts over heads, it necessarily captures model subcircuits at a much coarser level that provides a natural regularization against 'quirks' of any one dataset. Even if the dataset used in CHG has a spurious connection between John and even numbers, CHG will be unable to overfit to patterns that the LLM has not developed circuits for during its training. This exemplifies the utility of higher abstractions we noted in our previous response.
> > >
> > >     As a final note, our “red apples” example in the paper is intended to illustrate plausible incidental information that an LLM might encounter and represent, not a contrived or quirky pattern. By contrast, the “John → even number” scenario is a deliberately adversarial example. While we believe that CHG is likely to be robust to idiosyncratic artifacts such as these due to the head-level abstraction, we also acknowledge that all scientific tools remain sensitive to the dataset. Even a simple t-test or correlation measure will fail if applied in an inappropriate context, and it is ultimately the researcher’s responsibility to select methods judiciously. Our goal in presenting CHG is to provide fellow scientists and practitioners with a novel tool for identifying task-relevant model components with causal guarantees, which they can apply to the datasets and research questions most pertinent to their own work.
> > >
> > > 2. **Head roles across tasks:** We believe there is a key misunderstanding here. We are not claiming that individual head roles are inherently task-specific. Given the limited number of heads in LLMs (there are only 1024 heads in Llama 3.1 8B), this would not even be possible. Instead, what we mean is that while individual heads may have task-general functions, whole CHG head circuits (the conjunction of many heads) may be task-specific. That is, the combination of an LLM and a CHG matrix $G \in (0, 1)^d$ (where d is the number of heads) is likely to be task specific (note the difference in expressivity). The individual heads may well have task-invariant functions, which is precisely what our comparison to CMA in Section 3.3 helps identify.
> > >
> > >     We hope that this also clarifies the results we show in CCHG. First, we would like to reiterate our results from the paper. The characterization of CCHG as having “mostly failed” is not representative of the results we present in the paper. Our CCHG experiment successfully transferred in 7 out of 12 conditions, demonstrating that separability and cross-task generalization are achievable in a majority of tested scenarios. This is a non-trivial success rate in a non-trivial setting. Cross-task transfer effect is not an easy feat, nor is it a priori obvious that ICL and instruction-following subcircuits should be completely disjoint from the circuits involved in tasks like antonym recognition or country-capital queries.
> > >
> > >     Second, the very fact that head-level separability and cross-task transfer occur in multiple cases is the interesting result. It demonstrates that CHG can discover meaningful, partially invariant subcircuits, while also revealing that transformer models exhibit overlap, redundancy, and low modularity. We believe this nuanced outcome is more informative than an all-or-nothing notion of invariance and reflects the scientific insight CHG was designed to uncover.
> > >
> > > We hope this response clarifies the key misunderstandings and provides a fairer perspective on the results. If the reviewer finds these clarifications satisfactory, we respectfully request that the final assessment score be reconsidered to reflect them.

---

> > > > ### Comment · Reviewer_PMP1 · 2025-08-07
> > > >
> > > > Thanks for the detailed reply. I think my main concerns are not resolved and I look forward to your clarifications.
> > > >
> > > > 1. **Spurious correlations.**
> > > >
> > > > > “John -> even number” scenario is a deliberately adversarial example.
> > > >
> > > > But I see it as a simple illustration of a fundamental problem: correlation is not causation. Real-world data is full of such biases.
> > > >
> > > > You also claimed that "CHG will be unable to overfit to patterns that the LLM has not developed circuits for". I think this is a strong claim made without experimental evidence. A core issue remains that your method is designed to find what is predictive on a specific dataset, and that includes any spurious correlations within it.
> > > >
> > > > 2. **On the "causal discovery" claim.**
> > > >
> > > > The main problem is how the paper frames its contribution. In your reply to me, you narrow your claim to be about the causal link between "model components and the task" and agree it can't handle spurious correlations in the data. But in your discussion with Reviewer WNd7, you elevate your work to the level of "causal discovery".
> > > >
> > > > "Causal discovery" means finding a true graph from data up to some ambiguities. It is well-known that this is impossible with your type of data and model, without making much stronger assumptions, **even if your variables in the causal graph are gates/CHG matrix and task-specific losses**. This theoretical difficulty helps explain two key empirical findings in your paper: (1) why the CHG results are unstable across different random seeds, and (2) the mixed results of your CCHG generalization experiment, which I discuss next.
> > > >
> > > > 3. **Task invariance.**
> > > >
> > > > > “The individual heads may well have task-invariant functions, which is precisely what our comparison to CMA in Section 3.3 helps identify.”
> > > >
> > > > I think this is a misreading of your own results. That section's contribution is a method validation; it shows that for a single task, CHG and CMA find similar heads. It provides no evidence for task invariance because it never shows the same function working across different tasks.
> > > >
> > > > The lack of invariance is most clear in the CCHG experiment (Figure 5). You packaged a method for task-specific prediction into the grander framework of causal discovery. But when this claim was put to the test, the results were mixed and did not show strong generalization.
> > > >
> > > > Because of this overstatement and the lack of strong evidence for invariance, I will maintain my score.

---

> > > > > ### Author Response · Authors · 2025-08-08
> > > > >
> > > > > Thank you for the thoughtful discussion. If accepted, we will do our best to incorporate any clarifications and suggestions raised during this period into the final version.

---

### Official Review · Reviewer_WNd7 · 2025-07-02

**Clarity:** 3
**Significance:** 2
**Originality:** 1
**Rating:** 3
**Confidence:** 4

**Summary:**

This paper proposes Causal Head Gating (CHG), a scalable method for identifying the causal roles of attention heads in transformer models without relying on labels. By fitting soft gates to heads during next-token prediction, CHG classifies them as facilitating, interfering, or irrelevant to task performance. The method reveals that large language models contain multiple sparse, distributed sub-circuits and that functions like instruction following and in-context learning rely on separable mechanisms.

**Questions:**

- How does the choice of initialization for the gate matrix, random, all ones, or all zeros, affect the final learned mask?
 - What stopping criteria are used when training the gate matrix?
 - In Figure 1(b), what causes the abrupt bifurcation of the irrelevant heads around the 500th gradient update?
 - The manuscript refers to “causal scores” multiple times, does this term refer to the final gate values after training, or something else?
 - Lines 193–196 state, “We quantify these similarities…,” but the explanation was unclear to me. Could you please elaborate on what is being quantified and how?

**Ethical Concerns:**

["NO or VERY MINOR ethics concerns only"]

**Final Justification:**

While the authors have provided details regarding the differences of the proposed technique and similar existing methods in the literature, an empirical comparative analysis is still missing, which is essentially to assess a methodological contribution. Therefore, while promising, I cannot recommend to accept this work.

**Limitations:**

Due to optimization variability and the non-deterministic nature of head roles, multiple CHG runs may be needed to draw robust conclusions, diminishing its efficiency advantage over more straightforward methods like causal tracing and activation patching.

**Paper Formatting Concerns:**

None.

**Quality:**

2

**Strengths And Weaknesses:**

**Strengths**:
 - Unlike many existing mechanistic interpretability methods, the proposed CHG technique is both efficient and scalable. It effectively localizes groups of attention heads that collectively contribute to task performance, patterns that are often missed when applying causal interventions to individual components in isolation.
 - The paper validates CHG by triangulating its results through ablation studies and causal mediation analysis.
 - Furthermore, it uses CHG to demonstrate that the attention heads responsible for in-context learning and instruction following are distinct.

**Weaknesses**:
 - My primary concern with this work is its limited novelty. The proposed approach is quite similar to existing methods that use differentiable binary masks (DBMs) to identify model components encoding semantic information, particularly through counterfactual input pairs [1, 2].
 - Although the authors validate their method using ablation and causal mediation analysis, the paper lacks a comparative evaluation against established mechanistic interpretability techniques such as causal tracing, activation patching, and attribution patching.
 - The empirical finding that attention heads involved in in-context learning (ICL) and instruction following differ has already been suggested in prior work [3].
 - Moreover, I find the title somewhat misleading. It implies a broad focus on uncovering the roles of attention heads in various tasks, yet the paper largely sidesteps this issue. Only Section 4.4 addresses it directly, and even there, the emphasis is on distinguishing between ICL and instruction-following heads, rather than offering a general framework for identifying head-level functional roles.

[1] Davies et al, “Discovering Variable Binding Circuitry with Desiderata”, 2023.

[2] Prakash et al, “Fine-Tuning Enhances Existing Mechanisms: A Case Study on Entity Tracking”, 2024.

[3] Davidson et a “Do different prompting methods yield a common task representation in language models?”, 2025.

---

> ### Author Rebuttal · Authors · 2025-07-31
>
> We thank the reviewer for their thoughtful comments and provide the following clarifications and questions. While the rebuttal process does not allow for resubmitting revised content, we have aimed to address the reviewer’s concerns as clearly as possible and hope that our responses may help inform a reassessment of the work.
>
> 1.	Similarity to prior work – While Davies et al. and Prakash et al. use activation patching with manually constructed counterfactuals or desiderata, CHG learns gating parameters over all attention heads using only next-token prediction—avoiding the need for labeled tasks or handcrafted inputs. This results in a scalable, label-free method for causal attribution that differs substantially in both mechanism and scope. We are happy to cite these papers in the Related Work section, but we believe their goals and techniques are distinct from ours. Additionally, we note that none of the cited papers use the term “differentiable binary masks (DBMs).” Could you clarify what specific technique you had in mind by “DBMs,” and how it applies to these references or to CHG?
> 2.	Novelty of instruction vs. ICL – We appreciate the reviewer highlighting the work by Davidson et al. However, their preprint was first posted on May 17, 2025—two days after the NeurIPS submission deadline. We therefore view this as an instance of independent, concurrent discovery rather than prior work, and we hope this will be taken into account when assessing the novelty of our contribution.
> 3.	Comparison to other methods – We respectfully disagree with the assertion that we do not compare against established mechanistic interpretability techniques. Causal mediation analysis (CMA) is a structured form of activation patching, and both methods involve intervening on internal activations to identify causal roles. In Section 4.3, we benchmark CHG against CMA-based approaches and also replicate prior patching-style interventions.
> If there are specific forms of causal tracing or attribution patching you feel would strengthen the comparison, we would welcome your suggestions.
> 4.	Role of attention heads – We agree that CHG does not reveal the precise function of each head in isolation and is best used in combination with techniques like CMA. However, it provides a general, scalable framework for identifying which heads are causally relevant to which tasks—offering a simple, hypothesis-free tool for mapping attention head function across models and domains.
> 5.	Training details – As noted in the supplementary material (S3.1), we randomly initialize the gates. This helps uncover diverse configurations and avoids prematurely committing to a single solution. We train the gates for 500 gradient updates, which is sufficient for convergence in practice.
> 6.	Bifurcation in Figure 1b – As explained in Section S3.2, we first fit the initial gating matrix G, then switch to fitting G+ and G− after 500 updates. The bifurcation occurs at this point because the optimization objective changes—specifically, the regularization term becomes active (λ ≠ 0).
> 7.	Causal scores – Yes, the causal scores refer to the fitted gate values, as shown in Table 1. The score for a facilitating head is G−; for interfering, it is 1 − G+; and for irrelevant, it is G+ × (1 − G−).
> 8.	Similarity quantification – As described in the same paragraph, the similarity refers to the distribution of CHG values across different models on the same dataset. This is shown in Figure 3a, where cross-model correlations within a dataset reach 99.2%, supporting the observed alignment across plots in each row.
>
> We appreciate the reviewer comments and questions. If our responses to the reviewer concerns are satisfactory, we hope that you would reflect this in a higher overall rating for our work.

---

> > ### Comment · Reviewer_WNd7 · 2025-08-04
> >
> > Thank you for your responses. I look forward to the next round of replies.
> >
> > > Similarity to prior work – While Davies et al. and Prakash et al. use activation patching with manually…
> >
> > By methods that use a differentiable binary mask for localization, I was primarily referring to the following existing works, in addition to those already mentioned [1–4], none of which were cited or compared to the proposed technique. While Davies et al. and Prakash et al. use counterfactual interventions to train the mask, Yin et al. rely solely on the training data via next-token prediction. However, their method learns a vector for each head rather than a single parameter. Could you outline the similarities and differences between your approach and these existing methods to better situate your contributions?
> >
> > Additionally, could you describe the tasks or settings in which a researcher or practitioner might use the proposed technique without relying on counterfactuals? One potential use case is efficient fine-tuning, as suggested in [4]. Are there other scenarios where this technique would be particularly effective?
> >
> > > Comparison to other methods – We respectfully disagree with the assertion that we do not compare against…
> >
> > By not comparing the proposed technique with existing methods, I meant that no evaluation was performed on the following key metrics:
> > * The extent to which the localized model components differ between various methods, especially important given that the proposed technique is expected to identify causally relevant distributed components.
> > * Execution speed, particularly relevant for attribution patching, which is often promoted as a faster alternative to activation patching.
> >
> > Ideally, the authors could have replicated known circuits from the literature, such as IOI.
> >
> > [1] Cao et al, “How do Decisions Emerge across Layers in Neural Models? Interpretation with Differentiable Masking”, 2020.
> >
> > [2] Csordás et al, “Are Neural Nets Modular? Inspecting Functional Modularity Through Differentiable Weight Masks”, 2021.
> >
> > [3] Cal et al, “Sparse Interventions in Language Models with Differentiable Masking”, 2022.
> >
> > [4] Yin et al, “LoFiT: Localized Fine-tuning on LLM Representations”, 2024.

---

> > > ### Author Response · Authors · 2025-08-06
> > >
> > > Thank you for your response and your follow up points. Given the rebuttal rules that prohibit resubmission of any form, we will do our best to answer the reviewer concerns here and hope that our responses would help appeal for a higher reviewer score.
> > >
> > >
> > > 1. **Re – reliance on counterfactuals:** We emphasize that counterfactuals are only used in CCHG (Section 4.4); all other CHG analyses rely exclusively on standard next‑token prediction without any paired or handcrafted data. Even for CCHG, counterfactual pairs are optional rather than required—they simply help control extraneous factors and yield cleaner causal isolation.
> > > 2. **Re – applications:** Our primary objective is causal discovery for interpretability, rather than engineering performance‑optimized LLMs. Practitioners could apply CHG to tasks such as attention‑head pruning (e.g., Voita et al.) or localized fine‑tuning (e.g., Yin et al.), but our contribution is focused on revealing the emergent functional structure of LLMs, not on downstream optimization.
> > > 3. **Re – component differences across methods:** Section 4.3 of our paper directly addresses the overlap between CHG and CMA, showing that CMA-identified heads exhibit significantly higher CHG facilitation scores across both function vector tasks and symbolic reasoning tasks. This provides evidence that CHG reliably captures causally relevant distributed components and that its identified heads align with prior mechanistic findings. If the reviewer is referring to other gating-based methods (e.g., Yin et al. and Voita et al.), we direct them to our discussion of the theoretical differences between CHG and these papers (see discussion with Reviewer ymfF for Voita et al.), where we also explain how CHG addresses key limitations of these methods. While a broader cross‑method comparison could further characterize differences across techniques, our current goal is to validate CHG via convergent evidence, and we leave extended comparative analyses to future work.
> > > 4. **Re – execution speed:** As detailed in Appendix A.3, a single CHG run (1,500 gradient updates) completes in 15–60 minutes on one Nvidia H100, and a CCHG run (~500 updates) takes ~5 minutes. While we cannot perform full wall-clock comparisons with activation or attribution patching within the limited author–reviewer discussion period, we note that activation patching often requires multiple forward passes per head and scales poorly to distributed head analyses, whereas attribution patching is faster per intervention but requires many interventions to cover distributed components.
> > > 5. **Re: replicating the IOI circuit:** While we have not used CHG to specifically identify the IOI circuit, our experiment in Section 4.3 provides broad coverage by comparing our method across four different head types and seven tasks: the function vector heads evaluated on six tasks in Todd et al., and the symbol abstraction, symbol induction, and retrieval heads applied to the symbolic reasoning task in Yang et al. Moreover, we note that, like Todd et al. and Yang et al., Wang et al. (Ref. 25 in our paper) used CMA to identify the IOI circuit. Thus, while running an explicit IOI replication is beyond the short author–reviewer period, our existing comparisons to CMA give us high confidence that CHG would also recover this circuit.
> > > 6. **Re - related work:** We thank the reviewer for highlighting additional differentiable masking papers. Our intention was to focus on the most widely recognized and representative works in this space, such as Voita et al. (2019) and Li et al. (2021), which already apply differentiable gating to attention heads and were central to prior novelty discussions (e.g., by Reviewer ymfF). Voita et al. (2019) is particularly representative, with over 1,500 citations, and serves as a canonical reference for differentiable attention head gating. While we did not cite every variant, we believe the included works adequately contextualize our contribution, and we are happy to acknowledge the newly suggested papers in the final revision for completeness.

---

> > > > ### Author Response · Authors · 2025-08-06
> > > >
> > > > To fully address the reviewer’s concerns regarding novelty (from the original reviewer comments), we now discuss the four additional references in turn:
> > > >
> > > > 1. **Re - Yin et al.:** We acknowledge the similarities between our approach and Yin et al., as both rely on head-level scaling to identify components affecting task performance. However, the key distinction between Yin et al. and our work is the causal framing, which has substantial implications. Yin et al. use the scaling factor to identify heads with the largest performance changes, making their method best suited to detect what we refer to as interfering heads and a subset of facilitating heads, namely those most amenable to task performance improvement when modified. This means that the method in Yin et al. does not distinguish between irrelevant heads and facilitating heads that require little or no modification. In other words, a head that is already performing essential functions for a task appears indistinguishable from one that has no effect at all under their method. This distinction lies at the heart of causal discovery, which is the primary focus of our paper and is not covered by Yin et al.
> > > > 2. **Re - Cao et al. (2022):** While related, Cao et al. applies gating to hidden representations to LSTM, which operates at a different level of abstraction on a fundamentally different architecture. Applying gating at the head level allows asking different questions, such as how separable and modular are various functions in a transformer-based LLM. Moreover, we note that Cao et al. uses the Hard Concrete Distribution (also called the Gumbel-softmax), which has its own theoretic limitations that CHG does not (see our discussion with Reviewer ymfF for details).
> > > > 3. **Re - Cao et al. (2020):** While this paper aims to learn a causal mask, this mask is over the inputs rather than model components.
> > > > 4. **Re - Csordas et al.:** While this paper is similar in method in that it learns binary gates, it does so over all parameters in the model, which shares training complexity with the main model itself. Thus, it is unclear whether this method will scale to the level of LLM analysis, and indeed the paper uses a small transformer trained from scratch. Moreover, the use of the Gumbel softmax poses the same limitations as discussed above.
> > > >
> > > > In conclusion, even in light of these additional papers, none directly challenge the novelty of CHG. Our method uniquely offers a causal head‑level taxonomy (facilitating / interfering / irrelevant), can use both paired (contrastive) and unpaired (counterfactual-free) datasets, is scalable to modern LLMs, and does not share the same theoretic limitations of Gumbel-softmax based methods.
> > > >
> > > > We hope that our responses have fully addressed the reviewer’s concerns, and we would greatly appreciate the reviewer’s endorsement for the acceptance of our submission.

---

> > > > > ### Comment · Reviewer_WNd7 · 2025-08-07
> > > > >
> > > > > Thank you for the response. It has shed light on the differences between the proposed technique and similar existing methods in the literature. However, a comprehensive empirical comparison among these methods is still missing, which I believe is essential to assess a methodological contribution. Taking these factors into account, I have decided to raise my score to 3.

---

> > > > > > ### Author Response · Authors · 2025-08-08
> > > > > >
> > > > > > Thank you for the thoughtful discussion and the increased score. If accepted, we will do our best to incorporate any clarifications and suggestions raised during this period into the final version.

---

### Official Review · Reviewer_foUj · 2025-07-02

**Clarity:** 2
**Significance:** 3
**Originality:** 2
**Rating:** 5
**Confidence:** 3

**Summary:**

This paper introduces Causal Head Gating (CHG), a novel framework for evaluating the functional roles of attention heads in large language models (LLMs). Unlike prior approaches, CHG does not require hypothesis-driven datasets to analyze head behavior on specific tasks.

The method introduces gating coefficients, which control how much each attention head contributes to the model’s output. CHG is trained post hoc on frozen models using regularization to encourage sparsity, making it both interpretable and computationally efficient. By optimizing these gates, CHG segments heads into facilitating, interfering, or irrelevant categories.

The authors show that multi-head attention behaves as a flexible ensemble of overlapping sub-circuits, rather than as a collection of modular, context-independent components.

CHG is validated through multiple strategies conducted across a range of LLMs and tasks : the authors perform head ablations to test causal impact, replicate prior findings of Causal Mediation Analysis, a standard framework for estimating causal effect in neural network, and demonstrate strong correlation with CHG score and CMA-derived metrics.

Furthermore, the paper proposes Contrastive CHG, an extension that enables more fine-grained analysis by comparing head importance across different input conditions (e.g., retaining vs. forgetting context)

**Questions:**

How reproducible are the gating coefficients across different CHG runs with varied initializations or random seeds? Given this variability, what is a recommended number of CHG fits to achieve a robust or sufficiently consistent understanding of head roles for a given model and task? Do the authors observe convergence to similar gate configurations in tasks where head roles are assumed to be stable (e.g., math reasoning)?
Could they provide more insight into the nature or frequency of the divergence of G+ and G- matrices ? Are there specific tasks, model sizes, or layers where this divergence is more pronounced? Have they started to explore what these "richer head interactions" might entail, and how might CHG be extended to capture them?
Could the authors elaborate on the nature of the "overlapping sub-circuits"? Does it imply that different sets of heads can perform the same function (redundancy), or that a single head can contribute to multiple distinct functions depending on the context of other active heads? Can the authors speculate on the broader implications of this low modularity for future work in model compression, fine-tuning, or designing more interpretable architectures?
Would the authors consider extending their evaluation to include (i) coreference or long-context memory tasks (e.g., LAMBADA, WinoGrande), which are directly relevant to the retention/forgetting dynamics analyzed in Section 4.4, and (ii) instruction-following or chain-of-thought reasoning tasks (e.g., BBH, MMLU), where CHG may reveal distinct subcircuit behavior associated with in-context learning? Demonstrating CHG’s applicability in these settings, if feasible, would significantly strengthen the paper and would warrant a higher evaluation score. While the focus on LLaMA-3 models is reasonable for consistency, showing that CHG generalizes beyond a single model family would further reinforce its broader relevance.

**Ethical Concerns:**

["NO or VERY MINOR ethics concerns only"]

**Final Justification:**

After considering the rebuttal, I maintain that the paper presents a solid and timely contribution to interpretability in LLMs. The authors provided clear responses to most of my concerns, especially regarding the variability of CHG outputs, the interpretation of gating coefficients, and the justification for focusing on validated datasets. Their framing of CHG as a bootstrapping estimator clarifies its intended use and scope.

**Resolved issues:**
- Clarified the nature of variability across CHG runs and the intended interpretation of such variation.
- Justified the choice of datasets and models, framing them as grounded in prior validated findings.
- Provided insight into overlapping subcircuits and their implications for head interaction and modularity.

**Unresolved issues:**
- No new empirical evaluation on reasoning or memory tasks.

While these gaps limit the paper's breadth, they do not undermine the core contributions. The method is well-motivated, technically sound, and experimentally validated. Based on the rebuttal, I have increased my score, though I would not oppose a more conservative decision if other reviewers weigh the empirical limitations more heavily.

**Limitations:**

yes

**Quality:**

3

**Strengths And Weaknesses:**

Strengths :
Causal, Mechanistic Insight into Attention Heads
CHG offers causal rather than just correlational insight into attention head function, identifying heads as facilitating, interfering, or irrelevant. It reveals novel findings about Transformer internals, such as the presence of sparse, sufficient subcircuits, low modularity of head roles, and the separability of instruction following from in-context learning.


Scalable, General, and Practical Framework
CHG introduces only one learnable parameter per head, operates on frozen models, and fits in a small amount of time, even on billion-parameter LLMs. CHG also addresses limitations of previous methods like probing classifiers (which rely on labels) and CMA (which requires hand-crafted prompts and is less scalable) and applies directly to standard next-token prediction.


Robust Evaluation Across Models and Tasks
The framework is empirically validated across a diverse set of LLMs and tasks, demonstrating consistent segmentation of head roles.


Transparency and Reproducibility
The paper is methodologically rigorous, reporting t-tests and confidence intervals. The authors provide detailed algorithmic and experimental descriptions and plan to release the code publicly.

Weaknesses :
Limited Insight into Why Heads Matter or What They Compute
A significant limitation, as directly stated by the authors, is that CHG "does not identify why they matter or what they compute". While it effectively classifies heads based on their causal impact, it offers "limited insight into underlying mechanisms". This means CHG is best suited for "exploratory analysis alongside hypothesis-driven tools like CMA" rather than providing a complete functional explanation on its own—somewhat undermining its benefits from hypothesis-driven, annotated methods.


Occasional Divergence of G+ and G- Matrices
The authors note that, albeit rarely, the G+ (encourage retention) and G- (encourage removal) matrices can "diverge". This divergence "possibly reflect richer head interactions" that the current CHG framework might not fully capture. This points to a potential area where the model's complexity exceeds the current interpretive power of CHG.

Limited Dataset Coverage
While the authors evaluate CHG on a diverse set of tasks, the dataset selection omits coreference and memory retention tasks (e.g., WinoGrande, LAMBADA), which would have naturally extended the contrastive CHG experiments in Section 4.4. Additionally, instruction-following and chain-of-thought reasoning tasks (e.g., BBH, MMLU, GSM8K-CoT) are absent, despite CHG’s potential relevance to analyzing in-context learning and reasoning subcircuits. Testing on a reasoning-heavy dataset like BBH or MMLU would further strengthen the paper’s claims by extending CHG’s applicability to another important task class, but this omission does not undermine the core contributions.

Clarity Issues in Key Sections
Some parts of the paper, particularly Section 4.3, are densely written and conceptually compressed, making it difficult for readers unfamiliar with prior CMA work to follow the experimental setup and significance of the results.

While the paper lacks evaluation on reasoning and memory tasks, and CHG offers limited insight into what heads compute, these limitations do not necessarily undermine the core contributions. The framework is well-motivated and demonstrates potential for broader applicability.
I lean toward acceptance, as the paper offers a useful contribution to LLM interpretability, though I would defer to further discussion if others view the omissions as more fundamental. Additional experiments on reasoning tasks could strengthen the case and would warrant a higher score.

---

> ### Author Rebuttal · Authors · 2025-07-31
>
> We thank the reviewer for their thoughtful comments and provide the following clarifications and questions. While the rebuttal process does not allow for resubmitting revised content, we have aimed to address the reviewer’s concerns as clearly as possible and hope that our responses may help inform a reassessment of the work.
>
> 1.	Re: reproducibility across seeds – For each model and task, we report results from 10 different initializations, which often produce varying coefficients. This is illustrated in Figure 3b, where the “always” and “any” aggregation strategies reveal notably different patterns. However, we emphasize that consistency across seeds is not the objective. Variability in gating values reflects distributional variance, not irreproducibility. For example, if head A’s coefficient ranges from 0–1 across seeds, this suggests its effect is contingent on the configuration of other heads. In one run, head A may be suppressed because head B is not; in another, A may be active because head C is suppressed. This indicates that head A participates in a complex, interaction-dependent manifold rather than playing a strictly modular role.
> 2.	Re: recommended number of seeds – CHG is best viewed as a bootstrapping method for estimating a distribution over gating values, rather than identifying a single optimal configuration. Thus, the number of CHG fits depends on the model, task, and head. Some heads may have sufficiently modular or task-necessary functions that they are invariant to the initializations, whereas others have higher interactions with other heads. Either result may be interesting, depending on the practitioner’s goals. If the objective is to capture the distribution of causal scores for each head irrespective of their inter-head interactions, 20–30 fits is likely sufficient, as this is equivalent to estimating distributions bounded between 0 and 1. For broader analyses using min and max values as used in this work, 10 is likely sufficient.
> 3.	Re: convergence property – In mathematical reasoning tasks, we observe that task-relevant heads tend to concentrate in mid-layers, rather than early or late layers (see Figure 3b). Additionally, CHG reliably identifies heads also found by CMA (see Section 4.3), supporting the conclusion that these heads have stable, convergent roles.
> 4.	Re: divergence of G+ and G− – These divergences are rare: as shown in Figure 1c, only about 10 out of nearly 700 heads exhibit divergence. As for the precise nature of these cases, we have left this for future investigation, as they are fairly rare edge cases.
> 5.	Re: overlapping sub-circuits – Yes, both proposed interpretations are consistent with our view. The variation in CHG coefficients across seeds for some heads suggests that head A may be suppressed in one configuration because head B is not, and vice versa—i.e., A and B form an XOR-like interaction. This is likely to occur if A and B both serve similar roles, so that once A is considered “facilitating,” then B can be considered “irrelevant.” In such cases, task relevance should be interpreted jointly rather than independently. This suggests that even at the level of individual attention heads, LLMs may encode distributed representations rather than learning isolated “grandmother heads.”
> 6.	Re: broader implications for modularity – We are interested in further investigating modularity, particularly distinguishing between redundancy (where multiple heads serve the same function and some can be pruned) and contingency (where a head’s role shifts based on others being suppressed). The former could suggest certain heads can be pruned without much general performance loss. The latter could suggest that the behavior of an LLM could be changed by pruning
> 7.	Re: extension to other datasets – We are certainly interested in extending CHG to other datasets and domains! However, in this paper, we chose to place special focus on tasks and datasets with task-facilitating heads that have been confirmed by other papers (which is why we favored tasks from Todd et al. and Yang et al.) for the sake of validating our method before applying it to broader settings. We hope that the reviewer would appreciate the merit of our more conservative approach in favor of empirical grounding.
> 8.	Re: generalization to other LLMs – While we have run preliminary experiments with other LLM architectures and have observed results consistent with what we have reported here, we were unable to incorporate them into the full write-up of the present manuscript. We hope that the robustness across multiple domains and validation across the results of other papers would sufficiently demonstrate the generalizability of our approach.
>
> We appreciate the reviewer comments and questions. If our responses to the reviewer concerns are satisfactory, we hope that you would reflect this in a higher overall rating for our work.

---

> > ### Comment · Reviewer_foUj · 2025-08-01
> >
> > I appreciate the authors’ detailed and thoughtful responses. While some concerns remain, regarding dataset coverage, the rebuttal clarifies key points around variability, validation strategy, and broader applicability. Based on this, I am raising my score.

---

> > > ### Author Response · Authors · 2025-08-08
> > >
> > > Thank you for the thoughtful discussion and the increased score. If accepted, we will do our best to incorporate any clarifications and suggestions raised during this period into the final version.

---

### Note · Authors · 2025-08-12

We thank the reviewers for their constructive feedback. Three of the four reviewers explicitly noted raising their scores after our responses, indicating that our clarifications resolved their primary concerns. Below, we summarize the key points addressed during the discussion phase.

1. Novelty and comparison to existing work. Three reviewers (foUj, WNd7, ymfF) noted similarities between CHG and prior differentiable gating methods (e.g., Gumbel-based approaches). We provided theoretically grounded distinctions, including CHG’s ability to model head interdependencies and its causal taxonomy (facilitating / interfering / irrelevant). These arguments addressed limitations of prior work and were persuasive enough for all three reviewers to raise their scores.

2. Stability across seeds. Several reviewers (foUj, PMP1, ymfF) noted variability across seeds as a weakness. We clarified that this is intentional: CHG is highly scalable, and multiple runs reveal distributional variances that highlight redundancies and contingencies between heads. This distributional perspective prevents overinterpretation of any single mask and gives insight into the degrees of modularity in LLMs—an emergent property our method is designed to expose. We have also clarified that attention heads are unlikely to be perfectly modular with each other (with examples where the gating value of one head may affect another), so that the variance across seeds in CHG is not an indicator of instability of our method, but of the complexity of the underlying LLM our method is applied to.

3. Causality framing. One reviewer (PMP1) expressed confusion about our causal claims, suggesting that CHG may capture spurious dataset correlations. We have clarified that CHG does not identify causal relations between tokens in the data; rather, it identifies causal links between model components (attention heads) and model behavior on a task (output tokens). Moreover, because CHG directly proposes a counterfactual circuit to the default, unablated LLM and its effect on model outputs, this meets the basic definition of causality: the CHG manipulation changes model behavior.

We believe we have addressed all substantive reviewer concerns and clarified any remaining ambiguities, which mainly stem from possible misinterpretations rather than methodological shortcomings. We are confident that our contributions can be valuable to the community and look forward to the opportunity to share this work.

---

### Decision · Program_Chairs · 2025-09-17

**Decision:**

Accept (poster)

**Comment:**

This paper is a borderline. It introduces Causal Head Gating (CHG), a simple and scalable framework for interpreting the roles of attention heads by learning head-specific gates under interventional training. The method yields a useful taxonomy of facilitating, interfering, and irrelevant heads, validated through ablations and alignment with causal mediation analysis, and is further extended with a contrastive variant (CCHG) to differentiate circuits for in-context learning and instruction following.

Reviewers expressed mixed views: some concerns about novelty relative to existing differentiable masking and patching methods, stability across seeds, and over-claiming of “causal discovery.” However, others found the approach clean, practical, and well-validated, with two reviewers raising their scores after rebuttal. Overall, despite some limitations, I think this is a useful contribution to interpretability, providing a lightweight, label-free tool that researchers are likely to adopt. I recommend Accept, contingent on camera-ready clarifications about the scope of causality claims and broader comparisons to related methods.